# NovaChart: A Large-scale Dataset towards Chart Understanding and Generation of Multimodal Large Language Models

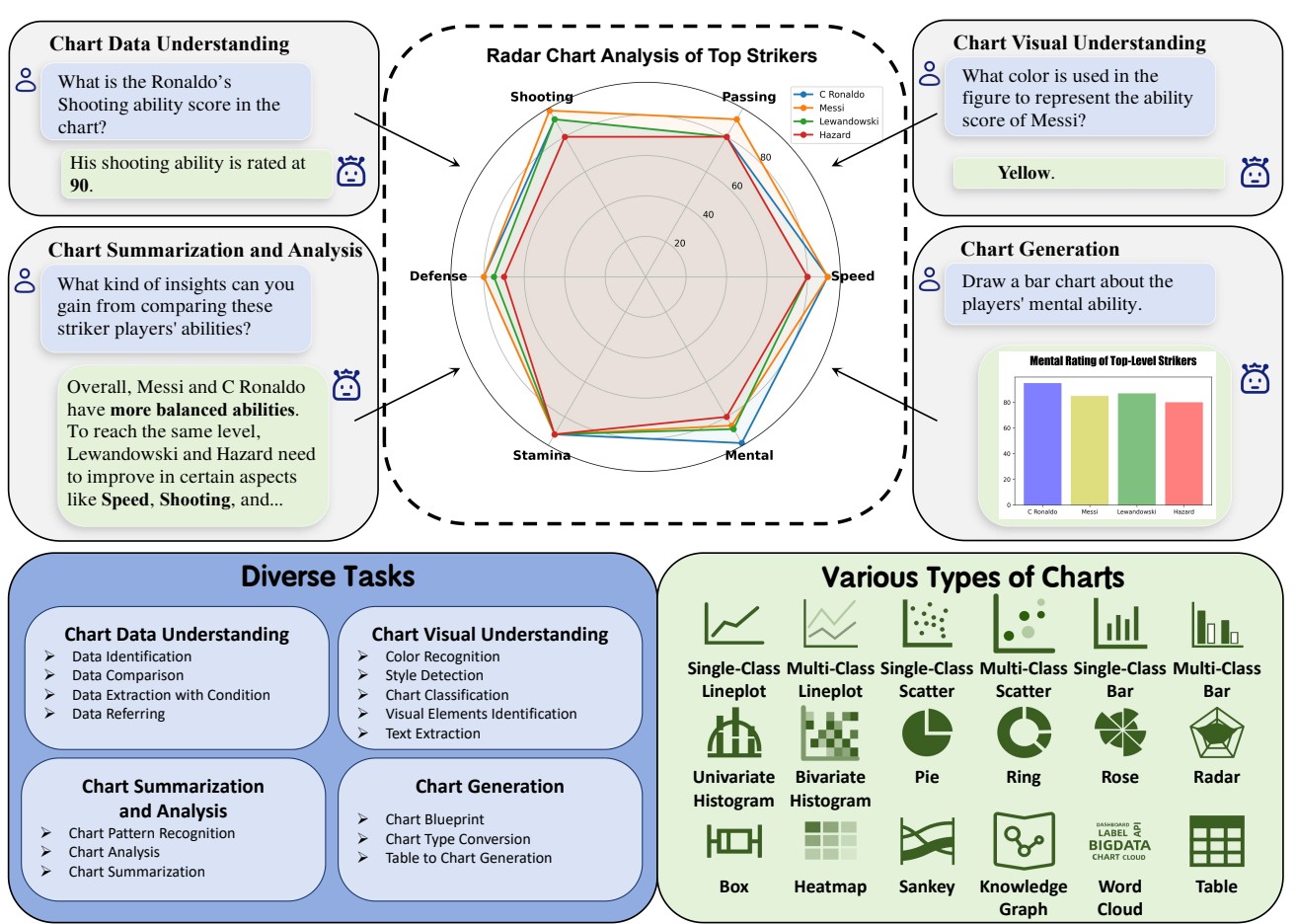

**Figure 1: The overview of NovaChart. NovaChart covers 15 chart understanding and generation tasks across 18 types of charts. We present demonstrations for some of the tasks at the top.**

## ABSTRACT

Multimodal Large Language Models (MLLMs) have shown significant potential for chart understanding and generation. However, they are still far from achieving the desired effectiveness in practical applications. This could be due to the limitations of the used training chart data. Existing chart datasets suffer from scarcity of chart types, limited coverage of tasks, and insufficient scalability, making them incapable of effectively enhancing the chart-related capabilities of MLLMs. To tackle these obstacles, we construct NovaChart, a large-scale dataset for chart understanding and generation of MLLMs. NovaChart contains 47K high-resolution chart images and 856K chart-related instructions, covering 18 different chart types and 15 unique tasks of chart understanding and generation. To build NovaChart, we propose a data generation engine for metadata curation, chart visualization and instruction formulation. Chart metadata in NovaChart contains detailed annotations, i.e., data points, visual elements, source data and the visualization code of every chart. This additional information endows NovaChart with considerable scalability, as it can facilitate the extension of

**Unpublished working draft. Not for distribution.**

chart instruction data to a larger scale and greater diversity. We utilize NovaChart to train several open-source MLLMs. Experimental results demonstrate NovaChart empowers MLLMs with stronger capabilities in 15 chart understanding and generation tasks by a large-margin (35.47%-619.47%), bringing them a step closer to smart chart assistants. Our dataset is now available at https://github.com/Elucidator-V/NovaChart.

## CCS CONCEPTS

• **Computing methodologies** → *Natural language processing*; *Computer vision*.

## KEYWORDS

Chart Understanding, Chart Generation, Multimodal Large Language Model

## 1 INTRODUCTION

Chart, the graphical representations that display data in a structured and organized manner, plays a pivotal role in information processing and application. In the current era of data explosion, charts intuitively represent the phenomena and insights encapsulated in data using abstract visual elements, which substantially boost the efficiency of information processing and representation, better-supporting decision-making for individuals or organizations. With the rapid development of Multimodal Large Language Models (MLLMs) [9, 19, 25, 28], researchers have recognized the promising prospects of applying this technology to intelligent chart understanding and generation, as MLLMs show strong abilities in understanding visual contents as well as following diverse human instructions [32]. Some pioneering works [8, 22, 30, 33] integrate chart-related instruction data from multiple sources and successfully cultivate better chart-related capabilities for MLLMs via fine-tuning. Despite recent research progress, existing chart MLLMs still face the following challenges existing in training data, thus far from meeting the requirements of real-world applications.

• **Scarcity of Chart Types:** The most critical issue of current chart datasets is the limited number of chart types. The majority of datasets [23, 24] only include 3 fundamental types of charts: bar charts, line charts, and pie charts. The existing dataset with the largest diversity [8] encompasses 10 types of charts, which remains distant from covering massive charts in real-world applications. Some widely used chart types in practical applications, such as histograms, radar charts, and word clouds, are commonly overlooked by existing datasets. Moreover, existing datasets [8, 11] suffer from an imbalance in the distribution of chart types. The proportion of bar charts even exceeds 80% in some widely used datasets [11] for training MLLMs. Insufficient training data for long-tail chart types can detrimentally affect the model's proficiency in effectively analyzing such charts in practical applications.

• **Restricted Task Diversity:** The diversity of instruction tasks plays a crucial role in the enhancement of chart understanding and generation capabilities of MLLMs. Additionally, increasing the diversity of supported tasks of MLLMs can also improve the user-friendliness of MLLMs as chart assistants. However, the majority of existing chart datasets mainly focus on restricted tasks with relatively fixed format and content, such as data extraction

[16], question answering [23] and summarization [11]. Some recent works [8, 30] have made efforts to consider some chart generation tasks like chart redrawing and text-to-chart. However, several more intricate tasks which closely aligned with practical applications have yet to be considered, such as conditional data extraction, visual elements recognition, and chart type conversion. Hence, existing datasets make it challenging for MLLMs to achieve the capability of task generalization, handling a wide range of chart-related tasks in real-world applications.

• **Limited Data Scalability:** The scalability of a vision-language dataset can be defined as its ability to support LLM-based instruction generation. Utilizing existing annotations as symbolic representations for visual data and leveraging language-only LLMs to generate customized instruction-following data involving various tasks is a common practice in visual instruction tuning for MLLMs [3, 14, 19]. Therefore, a crucial factor determining the scalability of a vision-language dataset is the richness of visual information encapsulated within the image annotations. For example, the scalability of COCO [15] is superior to CIFAR-10 [12], as the image caption in COCO provides a comprehensive description of the images, whereas CIFAR-10 only provides categorical information. However, the scalability of existing chart datasets is severely constrained. Few datasets [23] provide data points and a subset of visual elements (colors, etc.) within the charts. More annotations that could be utilized to support chart-related task expansion, such as source data and visualization code of charts, are notably absent, thereby limiting the scalability of these datasets.

In this paper, we propose NovaChart, a large-scale dataset for improving chart understanding and generation of MLLMs. NovaChart comprises 47K high-resolution chart images and 856K instances of chart-related instructions, encompassing 18 chart types and 15 distinct tasks with a balanced distribution. To build NovaChart, we develop a fully-fledged data generation engine consisting of four main steps: Raw data Acquisition, Data Curation, Image Styling and Visualization, and Instruction Formulation. The data engine empowers us to expand the number of chart types, produce chart images in diverse styles, and generate large-scale *chart instruction data* covering a wide range of chart understanding and generation tasks. Additionally, it provides extensive *chart metadata* including data points, visual elements, source data, and their visualization code, empowering NovaChart with superior scalability. Utilizing NovaChart, we fine-tune a series of open-source MLLMs, and achieve promising enhancements of performance on various chart-related tasks, making them one step closer to practical chart assistants. Furthermore, we also release several tools for the refinement and usage of our dataset, so that fellow researchers can conveniently utilize NovaChart to generate customized instruction data to enhance chart-related abilities for their personalized MLLMs.

**Contributions:** Our main contributions can be summarized as follows:

(1) A large-scale chart dataset for chart understanding and generation of MLLMs, with extensive coverage of chart types, various chart-related tasks, and good scalability.

(2) A fully-fledged data generation engine including Raw Data Acquisition, Data Curation, Image Styling and Visualization, and Instruction Formulation, supporting the construction of large-scale chart metadata and chart instruction data from scratch.

(3) A series of fine-tuned chart MLLMs with enhanced capabilities on 15 chart-related tasks, ranging from chart understanding to chart generation tasks.

(4) Several tools which can be employed for NovaChart extension, to help the development of customized MLLMs with specific chart understanding and generation capabilities.

## 2 RELATED WORK

### 2.1 Chart Datasets and Tasks

As mainstream tools for revealing the relationships, trends and patterns within complex data, charts, along with their associated tasks, have consistently been a focal topic in vision-language research. Early datasets primarily focus on chart understanding, and their corresponding tasks can be categorized as Chart Perception and Chart Cognition [30]. Chart Perception, such as data extraction [16], aims to extract the numerical and textual values of data points within the charts. Chart Cognition tasks, on the other hand, require a thorough understanding of chart content, including the relationships between data points, the overall trend of the data, and the phenomena it represents. For example, Chart QA [23, 24] requires models to answer free-form natural language questions about the chart, and Chart Summarization [11] targets a comprehensive summary of the chart. With the advancement of Multimodal Large Language Models, some chart instruction datasets [8, 17, 22] are proposed to equip MLLMs with the ability to tackle multiple chart-related tasks. These datasets integrate various existing chart understanding tasks and also encompass some novel generative tasks, such as chart redrawing and text-to-chart transformation. Chart datasets serve as the data foundation for training powerful artificial intelligence, which can assist humans in understanding and utilizing charts more efficiently.

### 2.2 Multimodal Large Language Models

By aligning visual features within Large Language Models, current MLLMs demonstrate impressive abilities in comprehending visual inputs and following human instructions [2, 19, 35]. They achieve expressive performance on several vision-language tasks, such as visual question answering [5, 7, 10, 21], image captioning [1, 15, 27], and visual grounding [34, 36]. However, chart-related tasks remain highly challenging for existing MLLMs, due to the absence of relevant training data [6, 31, 32]. To build MLLMs with better chart-related ability, two kinds of approaches are currently being adopted for training data construction. Some works [9, 33] integrate traditional chart datasets and tasks, such as PlotQA [24] and ChartQA [23] for question answering and Chart-to-text [11] for chart captioning, and convert them into unified instruction-following format for model training. For greater diversity, some other works [8, 17, 22, 29] utilize LLMs to generate novel chart instruction-tuning data covering more types of charts and multiple tasks. For instance, MMC [17] utilizes the caption of 7 types of chart as input for GPT-4, prompting it to generate instructions for chart information extraction and chart reasoning tasks. ChartLlama [8] employs GPT-4 to generate metadata for charts covering 10 types, and constructs associated instructions for 4 understanding and 3 generation tasks. However, training data of these works still encounter several challenges, including the scarcity of chart types,

insufficient diversity in tasks and obstacles to extending the current dataset to customized instruction-following data. These factors hinder the training of MLLMs with satisfactory capabilities of chart understanding and generation for real-world users.

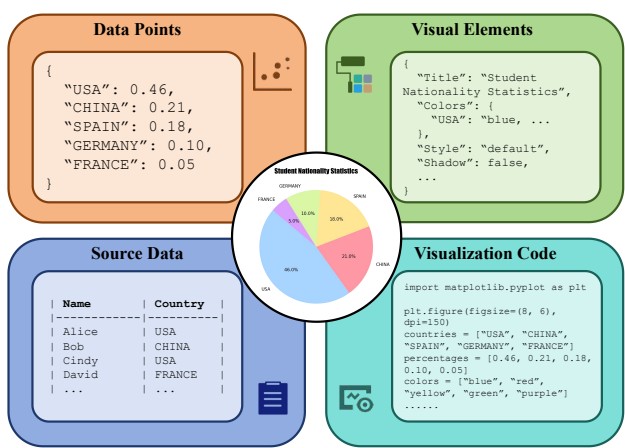

**Figure 2: Illustration of chart metadata in NovaChart.**

## 3 DATA OVERVIEW

NovaChart comprises 47K high-resolution charts and 856K chart instruction data distributed across 18 major types of chart and 15 unique tasks. NovaChart consists of two parts: detailed information of the chart itself, called *chart metadata*, and the multi-task instruction-response pairs, known as *chart instructions data*.

*Chart metadata.* As shown in Figure 2, for every instance of chart, we provide 4 kinds of annotations: 1) data points which are statistical units of information represented by numerical values on the chart's axes; 2) visual elements used in charts to convey information and enhance expressiveness, such as colors; 3) source data, the raw, unprocessed data samples from which the statistical chart is derived; 4) visualization code for chart images rendering with given data points and visual elements.

*Chart Instruction Data.* In NovaChart, the chart metadata contains abundant information, providing strong support for the construction of visual instruction-following data. We design a comprehensive set of 15 unique tasks, covering 4 kinds of tasks: 1) Chart Data Understanding, which aims to precisely understand the statistical data points within charts; 2) Chart Visual Understanding, which focuses on identifying particular visual elements in charts. 3) Chart Summarization and Analysis, which aims to summarize and analyze the phenomena behind the data. 4) Chart Generation, which focuses on generating executable visualization code (in Python) to help users create charts.

In general, NovaChart addresses the limitations of existing datasets, by enriching the types of charts, increasing the diversity of tasks and enhancing the scalability. NovaChart provides large-scale, high-quality data for chart understanding and generation.

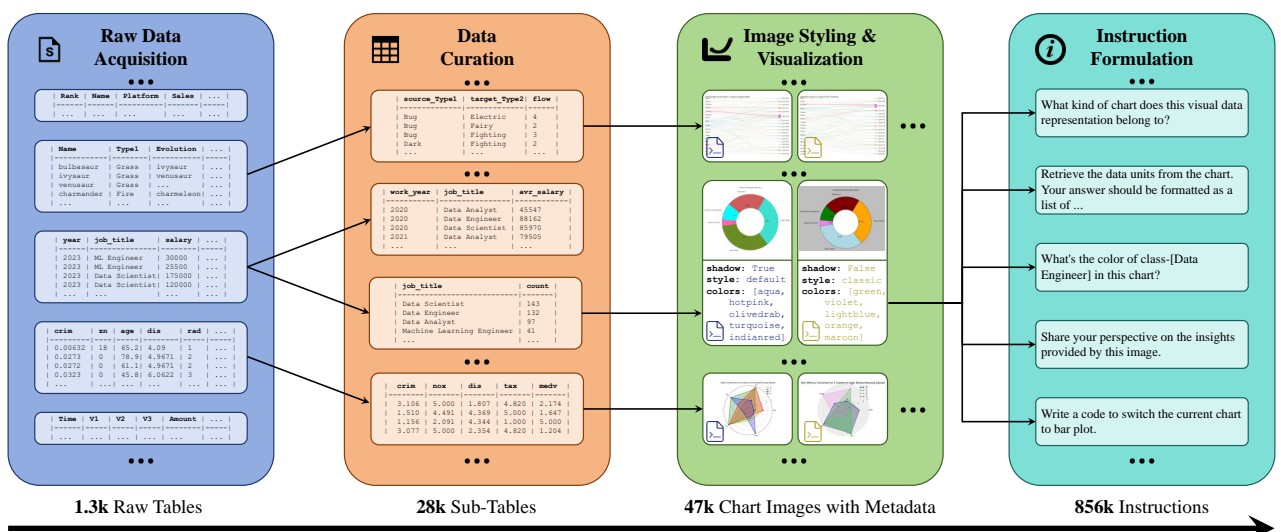

Figure 3: The overview of NovaChart data engine.

## 4 DATA ENGINE OF NOVACHART

The framework of the data generation engine of NovaChart is illustrated in Figure 3. It mainly comprises 4 steps.

1) Raw Data Acquisition: The first step is to collect extensive data from Kaggle and filter it to build the data foundation of NovaChart which are tables with real-world data values, covering diverse domains and themes.

2) Data Curation: In this step, we sample suitable attributes from the raw tables to get numerous sub-tables (source data) for deriving large-scale statistical charts. The data points in these charts are statistical values calculated based on the sub-tables.

3) Image Styling and Visualization: Based on the statistical data points, we employ various visualization tools and execute codes to render chart images with diverse visual styles. We now obtain the entire collection of metadata for each chart after this step.

4) Instruction Formulation: We design various chart understanding and generation tasks based on our chart metadata, and use LLMs to formulate instruction-following data of these tasks with good linguistic diversity.

In the following, we detail the construction of NovaChart.

### 4.1 Raw Data Acquisition

To build the foundational data for NovaChart, we collect relational tables with high user votes from Kaggle. Relational table is widely available on the Internet and is easily accessible. Its well-structured nature makes it an ideal raw data source for charts. Additionally, these Kaggle tables originate from the real-world scenarios of data science, encompassing a diverse range of themes and reflecting the data complexity and diversity in the real world. To further ensure the quality of raw data, we preprocess the collected relational tables as follows. We first remove non-English tables and the tables with too few (<50) rows. Then, for the remaining tables, we remove the columns without names, columns with an excessive number of missing values (>90%), and columns with too long contents (e.g., movie reviews). After preprocessing, we obtain a total of 1.3K tables with each table containing thousands of rows as the foundational raw data for NovaChart.

### 4.2 Data Curation

Data Curation aims to generate numerous source tables for deriving large-scale statistical charts. We first sample suitable attributes from the raw tables to obtain a large number of sub-tables (source data) that can be used for statistical analysis. We then calculate the chart statistics of interest based on the sub-tables and formulate the data points to be presented in charts.

To ensure that the charts are of practical value, it is important to select and sample suitable columns (attributes) from the raw tables. Following researches in data science [20, 26], we consider the following attribute types: Attributes with numerical values (named Numeric), e.g., "house price" and "salary". These attributes and their values can be used as the dependent variables in charts, such as the y-axis of line charts. As for non-duplicate numerical values (named Unique-Numeric), such as "year", they can also be valid independent variables (x-axis) in line charts; Attributes with string values and the number of unique values is no more than 5 (named Categorical), e.g., "gender". They can be used as the class label in charts that involve multi-class comparisons like multi-class bar charts. For the number of unique values no more than 25 (named Enumerable), e.g., "courses attended by student". They are suitable to be used as the qualitative variable for certain chart types, like bar charts and pie charts. We employ GPT-turbo-3.5 to classify the attributes in the raw tables. Specifically, we utilize in-context learning, providing the input prompt including the definitions of each attribute type (i.e., Numeric, Unique-Numeric, Categorical, and Enumerable), the total number of unique attribute values in the raw table, some example values of the attribute and the instruction

asking the model to determine which types the candidate attribute belongs to.

Then, we need to specify the suitable attribute types and attribute numbers to construct a source table for a specific type of chart. For example, the source table of a bar chart should contain one Enumerable attribute, and the source table of a line chart should contain one Numeric attribute and one Unique-Numeric attribute. The detailed descriptions of the suitable attribute types and attribute numbers for different types of charts are displayed in Appendix 1.3.

Next, for a certain chart type, we sample source tables (sub-tables) from every raw table to build this type of chart. At each time, we randomly select the required number of suitable attributes and 30-50 rows from the raw table. Note that, if we fail to sample a sub-table meeting the requirements from a raw table, we skip this raw table. After obtaining the sub-table, we calculate the chart statistics, and finally obtain the data points presented in the corresponding chart. For example, we calculate the occurrences of each Enumerable attribute value to build a bar chart. Finally, we get a large collection of 28K source sub-tables. Note that, one source sub-table can be used to generate multiple charts (with different visual styles) of the same type.

## 4.3 Image Styling and Visualization

After data curation, we visualize the statistical data points calculated from source sub-tables, obtaining corresponding chart images. We utilize mainstream Python libraries as tools for chart rendering, such as Matplotlib, Seaborn and Pyecharts. For better visual diversity, we randomly set visual elements (e.g., colors) in charts, and generate multiple images for a same instance of data points. Finally, we obtain 40K high-resolution chart images, along with the chart metadata including visual elements and visualization code.

## 4.4 Data Expansion

We implement several extensions for larger data scale and wider coverage of chart types. During data curation, we observe a shortage of sub-tables for line charts. Thus, we use data tables from Statista [11] as additional source data for generating line charts. We also utilize HTML/CSS to visualize some source sub-tables and regard these tables as an extra type of chart. Additionally, we manually collect some other commonly used charts in data visualization, such as knowledge graphs and word clouds. Note that, for knowledge graphs, the visualization code is unavailable. Through the aforementioned extensions, we finally get 47K charts along with their metadata, covering 18 unique chart types, as shown in Figure 1.

## 4.5 Instruction Formulation

We construct instruction-following data corresponding to our generated charts. Rich information contained in the chart metadata allows us to construct a wider variety of chart-related tasks. Our instructions involve various kinds of capabilities, from basic understandings of data points and visual elements, analysis of the phenomena behind charts, to generation of the charts. In total, we design 15 chart-related tasks and the definition of each task is presented in Appendix 1.2. In our design, we introduce several new tasks aligning with real-world scenarios. For example, we design

the task of data extraction conditioned on specific constraints. For chart visual understanding, we set some visual element identification tasks that are rarely considered in the past, such as determining whether the fitting curves in histograms exist. For chart generation, we propose novel chart-to-chart type transformation tasks. These designed novel tasks directly meet the diverse needs of different users and may enable the MLLMs with generalized chart capabilities through multi-task instruction tuning.

We employ different instruction formulation strategies to ensure the linguistic diversity of the instruction-response pairs. For the tasks whose correct responses can be manually obtained from the chart metadata, such as data extraction, visual element identification, and some chart generation tasks, we use GPT-4 to generate various instruction templates to transform them into instruction-following format, in order to enhance language diversity of instructions. For some more sophisticated tasks, such as chart summarization, we fully leverage the understanding and reasoning abilities of GPT-turbo-3.5 for chart instruction data generation. Specifically, we provide the GPT with the input prompt including the task instruction, in-context demonstrations, and the required current chart metadata. The output responses of the GPT as well as the corresponding instructions form our chart instruction-response pairs. Using the aforementioned methods, we ultimately formulate chart instruction data covering 15 diverse tasks, with a large scale of 856K instruction-response pairs.

## 4.6 Availability

Our NovaChart is now publicly available on https://github.com/Elucidator-V/NovaChart. In addition to the data resources, we provide three tools for fellow researchers to facilitate utilization and extension of NovaChart. Data curation tool enables users to re-initiate the process of obtaining chart metadata, enabling the generation of more chart instances of different topics. Chart visualization tool allows users to freely adjust relevant visualization parameters to generate chart images with more diversified visual styles. Instruction generation tool helps researchers leverage LLMs to create chart instruction data (based on chart metadata) covering a wider range of tasks, based on their own requirements. We aim to enable researchers to conveniently utilize NovaChart and assist them in generating high-quality chart data for their customized model training through these tools. We sincerely hope that our efforts can pave the way for the intelligent assistant with powerful capabilities in chart comprehension and generation.

## 5 EXPERIMENT

In this section, we reveal the characteristics of NovaChart by comparing them with other chart datasets. Furthermore, we fine-tune 3 open-source MLLMs using NovaChart and evaluate their performance on various chart-related tasks.

## 5.1 Dataset Characteristics

For comparison, we select a series of widely utilized chart datasets whose training splits have been publicly released. We consider task-specific chart datasets like PlotQA [24], ChartQA [23] and Chart-to-text [11] and Chart datasets covering mutliple tasks, such as SimChart9K [29], Unichart [22], MMC [17] and ChartLlama [8].

**Table 1: Comparison of NovaChart and existing chart datasets.**

| Dataset | Chart Type | Images | Instruction Data | Tasks | Chart Data Understanding | Chart Visual Understanding | Chart Summarization and Analysis | Chart Generation |
|---|---|---|---|---|---|---|---|---|
| ChartQA | 3 | 21.9K | 32.7K | 1 | ✓ | ✓ | ✓ | ✗ |
| PlotQA | 3 | 224K | 28M | 1 | ✓ | ✓ | ✓ | ✗ |
| Chart-to-text | 6 | 44K | 44K | 1 | ✓ | ✓ | ✓ | ✗ |
| SimChart9K | 3 | 9K | 9K | 3 | ✓ | ✗ | ✗ | ✗ |
| Unichart | 3 | 627K | 7M | 4 | ✓ | ✓ | ✓ | ✗ |
| MMC | 7 | 600K | 2.4M | 9 | ✓ | ✓ | ✓ | ✗ |
| ChartLlama | 10 | 11K | 160K | 7 | ✓ | ✗ | ✓ | ✓ |
| NovaChart | 18 | 47K | 856K | 15 | ✓ | ✓ | ✓ | ✓ |

**Table 2: Chart metadata of different datasets.**

| Dataset | Data Points | Visual Elements | Source Data | Visualization Code |
|---|---|---|---|---|
| ChartQA | ✓ | ✓ | ✗ | ✗ |
| PlotQA | ✓ | ✓ | ✗ | ✗ |
| Chart-to-text | ✓ | ✗ | ✗ | ✗ |
| SimChart9K | ✓ | ✗ | ✗ | ✗ |
| Unichart | ✓ | ✗ | ✗ | ✗ |
| MMC | ✓ | ✗ | ✗ | ✗ |
| ChartLlama | ✓ | ✗ | ✗ | ✗ |
| NovaChart | ✓ | ✓ | ✓ | ✓ |

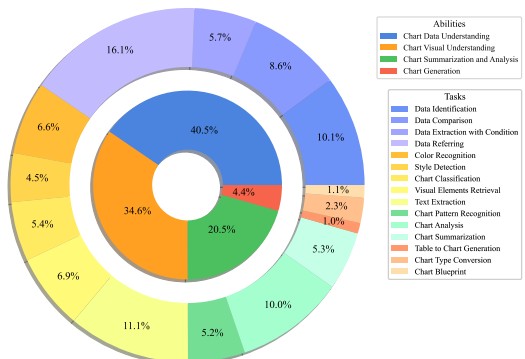

**Figure 5: Distribution of tasks in NovaChart.**

**Figure 4: Distribution of chart types in NovaChart.**

**Coverage of chart types**. As shown in Table 1, most existing training datasets focus on a limited range of chart types, like bar charts, line charts, and pie charts ChartLlama extends the number of chart types to 10 by directly using GPT-4 to generate data points and visualization code for charts, making it the state-of-the-art. However, our constructed NovaChart encompasses 18 types of charts (an increase 80% compared to ChartLlama). Moreover, NovaChart has a balanced distribution across different chart types as shown in Figure 4. The coverage of long-tail chart types contributes to a more comprehensive understanding and generation of charts.

**Diversity of chart-related tasks**. Compared to existing chart datasets, NovaChart encompasses a broader range of chart-related

tasks. Traditional chart datasets like ChartQA, PlotQA and Chart-to-text are constructed for a specific task. Unichart and MMC introduce various forms of understanding tasks such as chart information extraction and chart summarization, but fail to consider chart generation tasks. ChartLlama considers more tasks, including 1 task for data understanding, 3 tasks for chart summarization and analysis, and 3 tasks for chart generation. NovaChart, as shown in Figure 5, largely expands the variety of tasks on chart data understanding, chart visual understanding, chart summarization and analysis, and chart generation, and finally involves 15 distinct tasks. The diversity of tasks contributes to enhancing the generalization capability of MLLMs in various chart-related tasks, and thus enables MLLMs to satisfy different user requirements in real-world scenarios, making them more powerful and practical intelligent assistants.

**Data scalability**. Table 2 indicates that most existing datasets have limitations in terms of scalability, as they typically only provide the data points and some visual elements within charts. The absence of more detailed chart annotations hinders the datasets from suiting for various diversified and sophisticated tasks and instructions. To ensure good scalability, NovaChart provides comprehensive and detailed chart metadata including data points, visual elements, source table of the chart, and corresponding visualization code. These detailed metadata will facilitate the generation of more comprehensive and diverse instructions on chart understanding

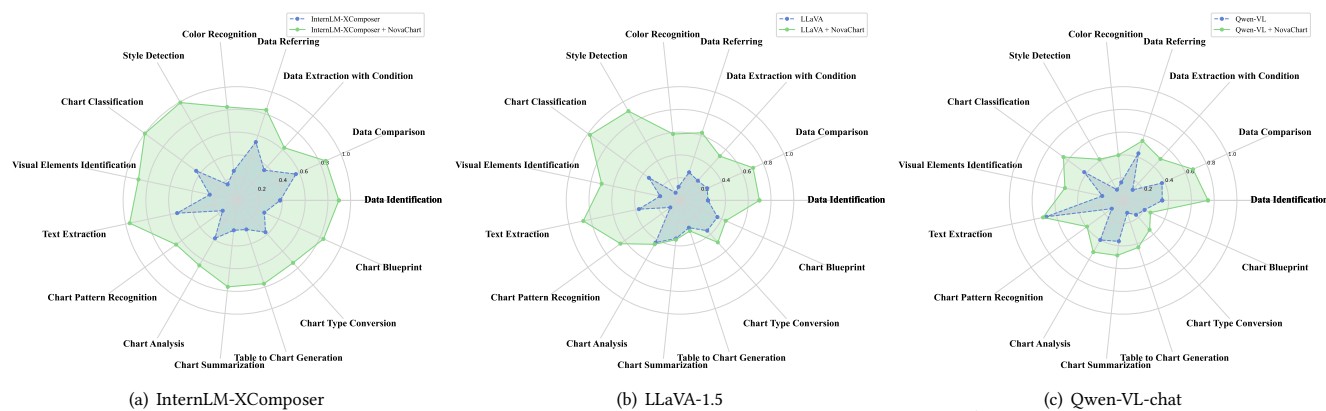

(a) InternLM-XComposer          (b) LLaVA-1.5          (c) Qwen-VL-chat

Figure 6: Comparison of model performance on 15 chart-related tasks before and after training.

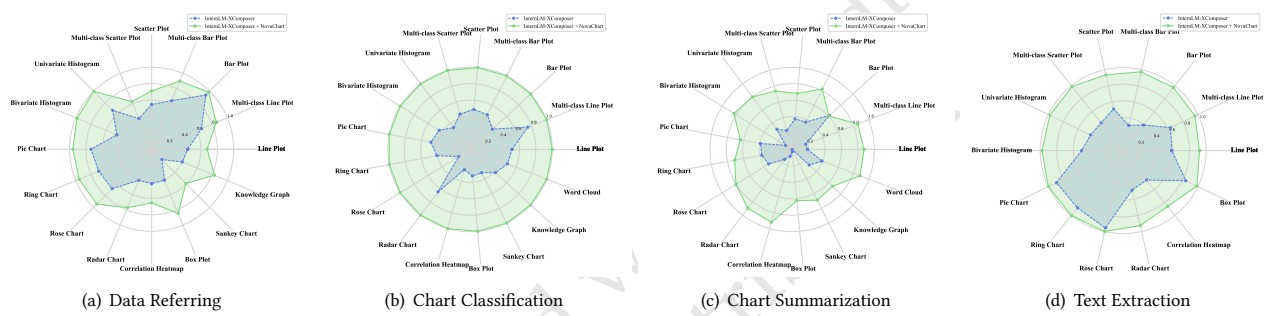

(a) Data Referring          (b) Chart Classification          (c) Chart Summarization          (d) Text Extraction

Figure 7: The performance of Internlm-XComposer on 4 typical tasks across different types of charts.

and generation, and further support the training of more powerful chart MLLMs.

## 5.2 Application

To validate the effectiveness of NovaChart in enhancing the chart understanding and generation capabilities of MLLMs, we fine-tune several open-source MLLMs and evaluate the model performance across a series of tasks.

### 5.2.1 Baselines and Evaluation Settings.
We fine-tune 3 representative open-source MLLMs with different architectures, LLaVA-v1.5 [18], InternLM-XComposer [35] and Qwen-VL-Chat [2] respectively. Hyper-parameter settings and more fine-tuning details are reported in Appendix 2.

We construct an independent evaluation set for NovaChart and evaluate models on all 15 chart-related tasks. As for evaluation metrics, we follow previous research and employ Exact Match (EM) for classification and question-answering tasks [30], and refer to RNSS [16] for numerical results in data referring tasks. For tasks requiring multiple data points extraction, we refer to the Levenshtein distance [13] and SCRM [29]. For open-ended chart summarization and analysis tasks, as well as chart generation tasks, we follow ChartLlama [8] and evaluate the performance using GPT-Score[4]. Detailed information on evaluation metrics is presented in Appendix 3.

### 5.2.2 Result Analysis.
We analyze the experimental results from two different perspectives: tasks and chart types.

**Results on different tasks.** Figure 6 presents the performance of 3 MLLMs on 15 chart understanding and generation tasks before and after fine-tuning on NovaChart. The detailed evaluation metric values of all tasks can be referred to in Appendix 4. We observe that before fine-tuning, MLLMs perform poorly across all tasks, particularly on chart generation. This indicates that although existing MLLMs possess certain capabilities of chart comprehension and generation in pre-training, they are still far from being applicable to real-world chart-related tasks. After fine-tuning on NovaChart, the MLLMs exhibit significant improvements on all tasks by a large-margin (35.47%-619.47%).The results suggest that by fine-tuning on NovaChart, models can acquire enhanced abilities in understanding of data points and visual elements, chart summarization, analysis, and generation. For some tasks, such as chart classification and text extraction, the fine-tuned models achieve excellent performance, demonstrating a remarkable ability to handle these tasks. This indicates the effectiveness and importance of our constructed NovaChart. At the same time, for some analysis and generation tasks, the improvements of the fine-tuned models are limited. This indicates that these tasks are highly challenging for existing MLLMs and should receive more attention in future research.

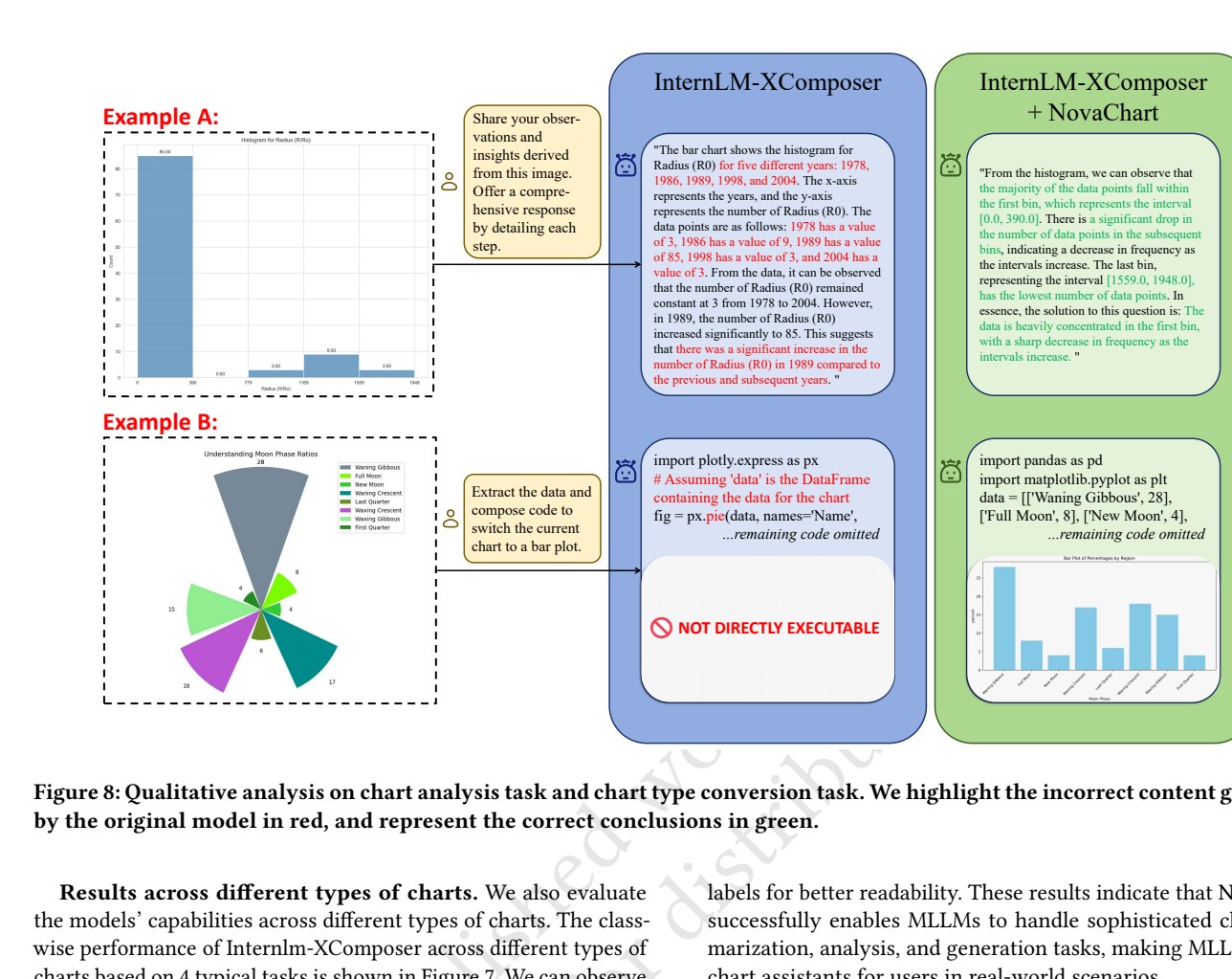

**Figure 8: Qualitative analysis on chart analysis task and chart type conversion task. We highlight the incorrect content generated by the original model in red, and represent the correct conclusions in green.**

**Results across different types of charts.** We also evaluate the models' capabilities across different types of charts. The class-wise performance of Internlm-XComposer across different types of charts based on 4 typical tasks is shown in Figure 7. We can observe that the model exhibits a general improvement for every type of chart. The model's performance score on the data referring task increases across all chart types, collectively resulting an improvement of 55.17% for this task. The GPT-score for the chart summarization task also improves 189%. As for the chart classification task, the model even achieves a remarkable 100% accuracy across all chart types. The results indicate that our NovaChart possesses a good balance across the distribution of different chart types, and thus enhances the MLLMs with better capabilities over all the chart types.

*5.2.3 Qualitative Results.* We present some examples of InternLM-XComposer on NovaChart for the tasks of chart analysis and chart type conversion in Figure 8. As shown by Example A, the original model (before fine-tuning) fails to correctly understand the meaning of chart axes, misinterprets the chart data, and thus draws incorrect conclusions. After fine-tuning, the model accurately interprets the histogram's data distribution and provides meaningful insights. In Example B, the original model cannot even understand user instruction on this task and thus generates incorrect codes, while the model after fine-tuning accurately extracts data points from the chart and generates executable visualization code. In addition, the model further considers certain details, such as rotating x-axis

labels for better readability. These results indicate that NovaChart successfully enables MLLMs to handle sophisticated chart summarization, analysis, and generation tasks, making MLLMs better chart assistants for users in real-world scenarios.

## 6 CONCLUSION AND FUTURE WORK

We present NovaChart, a large-scale dataset towards chart understanding and generation of MLLMs, consisting of 47K high-resolution charts for 18 chart types, detailed chart metadata, and 856k chart instructions covering 15 chart understanding and generation tasks. Compared with existing datasets, NovaChart demonstrates superior coverage of chart types, extensive diversity in tasks and good scalability, which eventually facilitates the enhancement of MLLMs' capabilities in chart-related tasks. We employ NovaChart to train several open-source MLLMs, and effectively improve their performance in chart understanding and generation. In addition to the publicly available data resources, we also provide a series of tools to facilitate the following researchers in extending our dataset towards customized chart MLLM construction.

In future work, we will continue to update NovaChart to provide better services for facilitating chart-related researches. We will explore more complex practical tasks, such as multi-chart QA that requires complicated reasoning on multiple charts. We sincerely hope the establishment of NovaChart will advance the development of MLLM-based intelligent chart assistants with exceptional capabilities in chart understanding and generation.

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
