# OpenReview forum: "NovaChart: A Large-scale Dataset towards Chart Understanding and Generation of Multimodal Large Language Models"
_acmmm.org/ACMMM/2024/Conference — MM2024 Oral_

### Official Review · Reviewer_gWNv · 2024-05-23

**Rating:** 4
**Confidence:** 3

**Summary:**

Current models for chart understanding and generation are limited by inadequate datasets that lack diversity and scalability. To address these issues, a new large-scale dataset called NovaChart has been developed. NovaChart includes 47,000 high-resolution chart images and 856,000 chart-related instructions, covering 18 chart types and 15 tasks. It features detailed metadata, such as data points, visual elements, source data, and visualization code, enhancing its scalability and utility. Experimental results show that NovaChart significantly improves the performance of models in chart-related tasks, advancing the development of smart chart assistants.

Thsi work is timely and relevant. I think papers like this will appear in the next couple of years for different domains.

**Strengths:**

This paper introduces a methodology and a chart dataset covering 18 types. The work specifically addresses the need for training domain-specific LLMs and also helps to train general-purpose LLMs with broader datasets. In contrast, the state-of-the-art method covers 10 charts.

The need for such a work is explored in detail.

This is a very focused paper on building datasets for a specific category (chart type) and demonstrating its effectiveness on other models by incorporating them.

**Limitations:**

"Data Curation aims to generate numerous source tables for deriving 437 large-scale statistical charts. "  section 4.2 - It isn't clear to me whether you manually created the sub-tasks or automatically.

The novelty of the work is the methodology and the dataset created - however, not so much in technical details.

**Suitability:**

2

---

### Official Review · Reviewer_whjV · 2024-05-25

**Rating:** 3
**Confidence:** 3

**Summary:**

Existing chart datasets suffer from limitations like scarcity of chart types, limited coverage of tasks, and insufficient scalability, which hinders the progress of MLLMs in chart-related capabilities. To address these issues, the authors built NovaChart, which contains 47K high-resolution chart images and 856K chart-related instructions, covering 18 different chart types and 15 unique tasks of chart understanding and generation. NovaChart includes detailed annotations like data points, visual elements, source data, and visualization code for each chart, which provides considerable scalability to extend the instruction data. The authors utilized NovaChart to train several open-source MLLMs, and the experimental results show significant improvements (35.47%-619.47%) in 15 chart understanding and generation tasks, bringing the models closer to practical smart chart assistants.

**Strengths:**

The key novelty of this work is the construction of the large-scale NovaChart dataset, which addresses the limitations of existing chart datasets. NovaChart covers a much wider range of chart types and chart-related tasks compared to prior datasets. The paper provides a clear and thorough description of the data generation engine used to create NovaChart, including metadata curation, chart visualization, and instruction formulation.   This technical approach seems sound and well-designed. The authors also explain the dataset's scalability advantages due to the detailed annotations (data points, visual elements, source data, visualization code) for each chart.

The authors demonstrate the usefulness of NovaChart by training several open-source MLLMs on the dataset and evaluating their performance on 15 chart understanding and generation tasks.  The significant performance improvements (35.47%-619.47%) observed across these tasks provide strong empirical evidence for the value of the NovaChart dataset. Comparing the results to previous methods would further strengthen the evaluation, but the authors still present a compelling case for the effectiveness of their approach. The paper is well-structured and easy to follow, with clear explanations of the dataset, the tasks, and the experimental setup and results.

**Limitations:**

- While the authors demonstrate significant performance improvements on the 15 chart-related tasks when using NovaChart, they do not provide detailed comparisons to the performance of previous state-of-the-art methods or datasets.
- The authors do not discuss potential biases or skews that may exist in the NovaChart dataset, such as overrepresentation of certain chart types, data domains, or visualization styles. Understanding and addressing such biases is important to ensure the fairness and robustness of models trained on the dataset.
- The paper does not include any ablation studies to investigate the individual contributions of the different components of the NovaChart dataset (e.g., the impact of the detailed annotations). Such ablation analyses could provide deeper insights into the key factors driving the performance improvements.

**Suitability:**

2

---

### Official Review · Reviewer_36Cj · 2024-05-25

**Rating:** 4
**Confidence:** 4

**Summary:**

This paper introduces NovaChart, a comprehensive dataset designed to enhance the chart-related capabilities of Multimodal Large Language Models (MLLMs). NovaChart includes 47,000 high-resolution chart images and 856,000 chart-related instructions, covering 18 chart types and 15 unique tasks. The dataset aims to address limitations in existing chart datasets, such as limited chart types, task diversity, and scalability. NovaChart's metadata includes detailed annotations like data points, visual elements, source data, and visualization code, which facilitate extensive scalability. Experimental results demonstrate that MLLMs trained on NovaChart show significant improvements across various chart understanding and generation tasks. The dataset and accompanying tools are publicly available to support further research and development in the field.

**Strengths:**

1. NovaChart provides a large-scale dataset with 47,000 high-resolution charts and 856,000 chart-related instructions. This extensive coverage includes 18 chart types and 15 distinct tasks, addressing the scarcity and diversity issues of existing datasets.
2. The dataset and tools are made publicly available, promoting transparency and enabling further research and development in chart understanding and generation.
3. The paper provides several tools to facilitate the extension and customization of the dataset, enabling researchers to generate more chart instances and create tailored instruction data for specific needs.

**Limitations:**

1. The dataset relies heavily on tables sourced from Kaggle and other online platforms, which may introduce biases based on the nature and quality of the available data
2. The quality of the generated instruction-following data and the overall performance improvement is somewhat dependent on the capabilities of the large language models (e.g., GPT-4) used for instruction generation.
3. missing reference: LRV-Instruction (ICLR'24)

**Suitability:**

3

---

### Official Review · Reviewer_sZCx · 2024-05-27

**Rating:** 4
**Confidence:** 4

**Summary:**

The paper introduces NovaChart, a large-scale dataset designed to enhance the chart understanding and generation capabilities of Multimodal Large Language Models (MLLMs). NovaChart includes 47,000 high-resolution chart images and 856,000 chart-related instructions, covering 18 different chart types and 15 unique tasks. The dataset is accompanied by a data generation engine for metadata curation, chart visualization, and instruction formulation. The authors demonstrate that training MLLMs with NovaChart significantly improves their performance on various chart-related tasks.

**Strengths:**

- The data generation engine's systematic process for creating detailed chart metadata and instruction-following data showcases a robust methodology.

- The potential applications of NovaChart in developing intelligent chart assistants are significant, offering practical value to both academia and industry.

**Limitations:**

- While the dataset is publicly available, the paper lacks detailed information on the reproducibility of the data generation process. More specifics on the exact parameters and settings used in the generation engine would help in verifying and replicating the results.

- The paper does not provide sufficient details on the criteria and methods used for selecting and filtering the raw data from Kaggle. Specific examples or guidelines on how attributes were classified and chosen for inclusion in the dataset would enhance clarity and reproducibility.

- While the paper mentions removing columns with over 90% missing values, it lacks an explanation of how other missing values were handled. Different strategies for imputation or removal can significantly impact the quality and representativeness of the dataset.

- The paper states that GPT-4 was used to generate instruction templates, but it does not detail the prompts or in-context learning strategies employed. This lack of detail makes it difficult to evaluate the quality and variety of the instructions generated.

- The fine-tuning process for the three open-source MLLMs (LLaVA-v1.5, InternLM-XComposer, and Qwen-VL-Chat) is briefly mentioned, but the specific hyperparameter settings, training schedules, and computational resources used are not detailed. This information is essential for reproducibility and understanding the feasibility of using NovaChart.

- The paper fails to cite several key prior works that have made significant contributions to chart understanding and generation. These include:

Cheng, Z.-Q., Dai, Q., Li, S., Sun, J., Mitamura, T., & Hauptmann, A. G. (2023). ChartReader: A unified framework for chart derendering and comprehension without heuristic rules. In *Proceedings of the IEEE/CVF International Conference on Computer Vision*.

**Suitability:**

2

---

### Meta-Review · Area_Chair_CwCV · 2024-07-01

**Recommendation:** Accept (Oral)
**Confidence:** 5

**Metareview:**

The only reviewer "whjV" changed their decision to "Borderline Accept", which means that all four reviewers' decisions were on the "Accept" side.
Therefore, this paper should clearly be judged as "Accept".